# Mod3D: A low-cost, flexible modular system of live-cell microscopy chambers and holders

**Siobhan Goss**[1☯], **Carlos Barba Bazan**[1☯], **Kaitlyn Neuman**[1], **Christina Peng**[1], **Nola Begeja**[1], **Celeste Elisabeth Suart**[1], **Ray Truant**[1,2]*

1 Department of Biochemistry and Biomedical Sciences, McMaster University, Hamilton, Ontario, Canada
2 Center for Advanced Light Microscopy (CALM), McMaster University, Hamilton, Ontario, Canada

☯ These authors contributed equally to this work.
* truantr@mcmaster.ca

**Data Availability Statement:** All STL files are available from NIH 3D print Exchange server ID: 3DPX-016293. https://3dprint.nih.gov/discover/3dpx-016293.

## Abstract

Live-cell microscopy imaging typically involves the use of high-quality glass-bottom chambers that allow cell culture, gaseous buffer exchange and optical properties suitable for microscopy applications. However, commercial sources of these chambers can add significant annual costs to cell biology laboratories. Consumer products in three-dimensional printing technology, for both Filament Deposition Modeling (FDM) and Masked Stereo Lithography (MSLA), have resulted in more biomedical research labs adopting the use of these devices for prototyping and manufacturing of lab plastic-based items, but rarely consumables. Here we describe a modular, live-cell chamber with multiple design options that can be mixed per experiment. Single reusable carriers and the use of biodegradable plastics, in a hybrid of FDM and MSLA manufacturing methods, reduce plastic waste. The system is easy to adapt to bespoke designs, with concept-to-prototype in a single day, offers significant cost savings to the users over commercial sources, and no loss in dimensional quality or reliability.

## Introduction

Live-cell microscopy imaging has evolved from single chamber observations to large-scale uses in High Content Analysis. Commercial live-cell imaging chambers and plates typically involve injection molded thermoplastic bonded to the glass. However, these chambers are typically expensive relative to 6-, 12-, and 96-well plate formats and plastic dishes, often by an order of magnitude or more, due to the added complexity of a bonded glass surface. Strict manufacturing standards ensure the dishes are flat and are reliably constructed to hold media without leaking. Assembled live-cell chambers are available from commercial sources, but are machined in metal or plastic and are typically very expensive and limited in the scope of use, but are a common choice for perfusion experiments. However, in a typical cell biology laboratory, these live cell chambers can add significant costs.

3D printing methods to manufacture plastic-based items have been historically reliant on expensive commercial industrial printers protected by a series of hardware and software patents. From 2015, open-source software-driven kits and complete printers cut the costs of these

**Funding:** This study was supported by the Natural Sciences and Engineering Research Council of Canada through a grant awarded to RT (RGPIN-2020-06642). The funders had no role in study design, data collection and analysis, decision to publish, or preparation of the manuscript.

**Competing interests:** The authors have declared that no competing interests exist.

devices drastically. This initiative was started at the University of Bath as the replicating rapid prototyper (RepRap) project in 2005 [1], which was released publicly as an open, free license and GNU general public license. This open-source hardware design was coupled to open-sourced Marlin firmware which could translate 3D modeled shapes into G-code machine commands to direct the tool movements in 3D printers. Marlin runs on inexpensive 8-bit Atmel microcontroller chips at the center of the open-source Arduino/Genuino platform [2]. Most early designs were filament deposition modeling, or FDM, in which a plastic filament is forced into a heated extruder at the glass-transition melting point of the material to layer patterns in a Cartesian XYZ space. Mixed orientations of the layer depositions give models high strength in multiple directions of force and are typically flat within 50 micrometers with print resolutions at 100 micrometers or higher, depending on extrusion nozzle size.

Another type of consumer 3D printing, stereolithography (SLA), does not use filaments nor heat, but rather UV-crosslinked resin layers in patterns directed by a <405nm laser line controlled by a mirror and galvanometer. This was further derived to Masked Stereo Lithography (MSLA) by the use of inexpensive light-emitting diode (LED) arrays and consumer electronics-sourced liquid crystal displays (LCD) to replace the galvanometer-directed laser light with a lithography mask created by the LCD in front of an array of 405 nm LEDs. A platform controlled by a stepper motor can then provide resolutions in Z at 25 micrometers, with an Y axis resolution of 1.25 micrometers. For both FDM and MSLA consumer printers, costs have been reduced to under US$300. Thus, subsequent adoption to research labs is becoming increasingly popular for the use of small research-based plastic objects such as racks, holders, brackets, and protein models, but typically not replacing plastic consumables. The cost of scientific laboratory use of injection molded consumable plastics is typically driven by high costs of tool and die manufacturing and the limited markets of potential recovery of those costs. However, these factors are not affected by 3D printing.

Given the resolution and precision of 3D printing, as well as the inherent ability to customize, applications in microscopy are numerous to develop low cost specialized solutions [3, 4] to open sourced projects that significantly lower the cost barrier to high performance microscopy, such as the OpenFlexure system [5–7]. 3D printing has been adopted successfully to create microscopy stage chambers for the study of model organisms in three and four dimensions [8]. Molds have been designed for custom production of silicone-based live cell chambers for multimode and multiplex observations [9]. Given the introduction of these printers to the market, 3D printing is quickly becoming essential to the cell biology or teaching laboratory.

Here we sought to manufacture a lab consumable: live-cell microscopy incubation chambers manufactured by MSLA, bonded to standard 22x22mm glass coverslips, held within universal 96-well format platforms manufactured by FDM, self-assembled with magnets, and designed to be reused. We set out to achieve ease of manufacture, low cost, ability to modify for bespoke applications, reliable containment of media; high dimensional quality, a portion of reusability and reduced environmental impact through the use of minimal/biodegradable plastic. The designs were evolved to be modular, allowing a mix of well sizes and types as opposed to one set format with unused wells. Each design can be used within a single universal 96-well format stage holder, minimizing waste and maximizing experimental flexibility.

## Results

The challenges of this project included many (>100) empirical attempts of designs, first using just FDM printing. We attempted to use both polylactic acid (PLA) as well as Polyethylene Terephthalate Glycol (PETG) filaments for 3D printing. We found this method unreliable, with the resolution not up to the standards required, a poor ability of FDM prints to be

watertight, and a glass bonding surface with too many potential imperfections to allow reliable bonding. The printing time was also several hours by printing chamber replicates. However, we did find the use of FDM superior for the chamber holder, which we designed on a universal 96-well size format at 4mm thickness with a 20% grid infill to provide a very strong, flat and stiff platform that did not expand or change dimensions up to 37°C (Fig 1A). We settled on a final design that could hold up to three 22x50mm chambers or six 22x22mm chambers (Fig 1B), with the overall dimensions of a universal plastic 96-well dish and a lid made of a frame of FDM plastic bonded to a cut sheet of 0.2mm polystyrene (Fig 1C–1E). Within the top and bottom halves of the holder, two or four 5x1mm neodymium magnets on each half were used to self-assemble the lid onto the base (Fig 1C–1E) and hold the chambers in place to prevent floating under objective pressure while using water or oil immersion objectives. Directional arrows, a logo, and one chamfered corner were incorporated into the design to aid with orientation. Each half of the chamber holders can take 2–2.5hrs to print at standard settings on the printer (200 micrometer layers), but these holders and lids can be reused with a sterilization cycle indefinitely.

For the chambers, we switched from using FDM to printing with MSLA resin. This solved two problems: the MSLA prints are essentially one piece of plastic, so leaking was never observed, and the top final surface of the print was extremely high quality for glass bonding. We took advantage of the high print resolution to emboss an alphanumeric labeling of chambers onto the sides to keep track of well identities during experiments. The base design was a 22x50mm chamber that fit two standard 22x22mm #1.5 glass coverslips, with a bonding

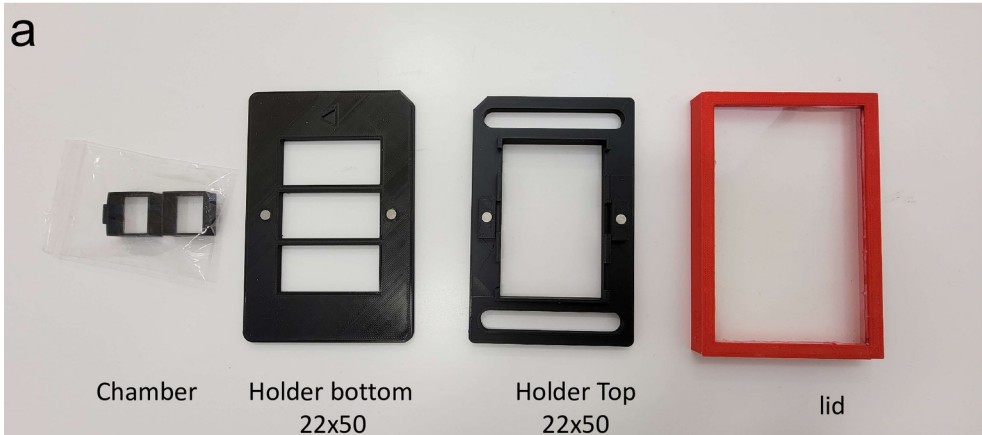
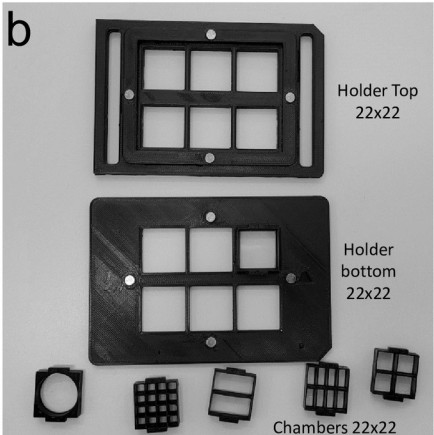
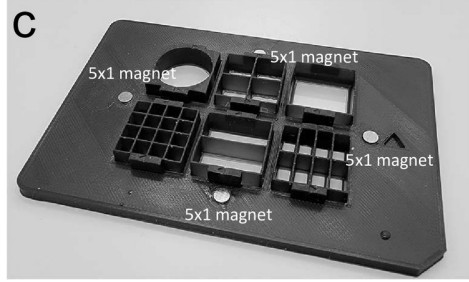
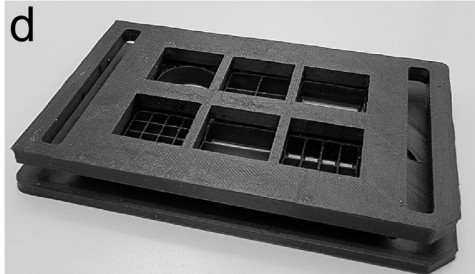
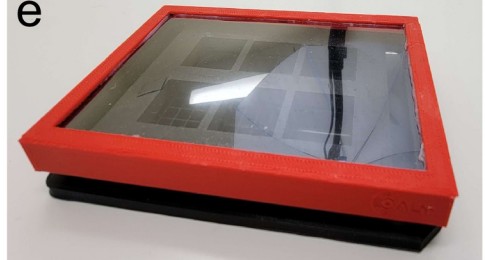

**Fig 1. Mod3D live-cell microscopy system.** (a) Examples of 22x50 mm 3-chamber holders in the universal 96-well plate format, with MSLA printed chambers, FDM printed holder, inserted magnets and reusable PLA/polystyrene lid. (b) Examples of the 6-chamber holders with different chamber designs. (c-e), Modularity and assembly of the system, where the holders and the lids are reusable.

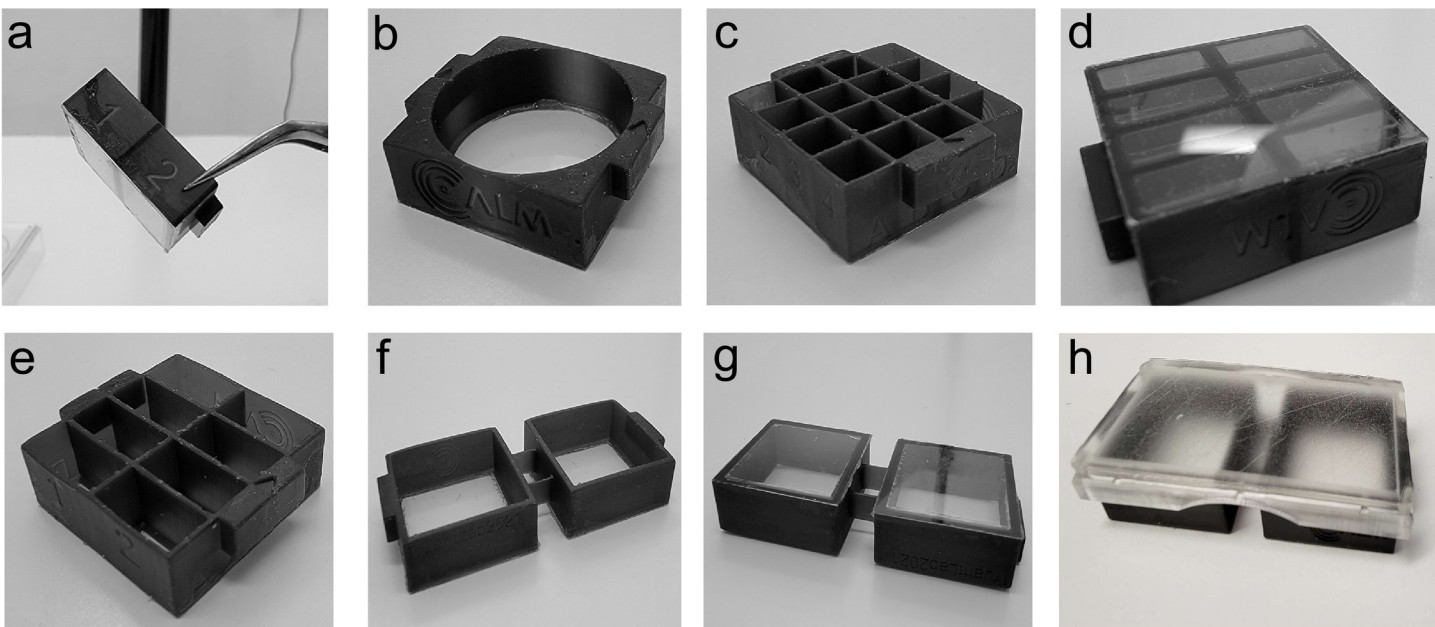

**Fig 2. Examples of varieties of chamber designs in Mod3D.** (a,b) Examples of embossed alphanumeric labels on 4-well (750 μl per well) or 16-well (145 μl per well) chambers. (c) Round single well design (2.8 ml). (d) Example of glass bonding using RTV silicone at the bottom of the chambers. (e) Example of 8-well chambers (260 μl per well). (f) Bridged 2-well chambers in 22x50 mm format, shown bonded to 22x22 mm glass coverslips in (g). (h) Bridged 2-well chambers bonded to a 22x50 mm glass coverslip, with reusable plastic lid.

surface of 1.5mm wide edges, ensuring a reliable bond between the plastic and glass. Within this base template (Fig 2A–2E), we were then able to design multiwell plates up to 32 wells per chamber, or 96 wells total within a single three-position carrier. Chambers were designed so that a 150 micrometer coverslip (#1.5) sat 100 micrometers below the plastic holder to prevent any friction between objective lenses and holder plastic during XY travel.

For bonding the coverslips to the plastic chamber, we previously successfully used room temperature vulcanization (RTV) silicone (Silicone I, General Electric) on manufactured plates in which we milled a hole on the bottom of a 35-mm tissue culture dish [10]. RTV silicone has excellent glass bonding properties, is resistant to all solvents used in cell fixation and permeabilization and can withstand temperatures from -60˚C to 200˚C. We found two-component silicones cured too quickly for practical handling and Vinyl-terminated Polydimethylsiloxane (PDMS) (Silguard, Dow Chemical) had poor adhesive properties or required priming steps. RTV silicones can be cured by the release of acetic acid or oxime-based curing. Both curing compounds can be toxic to cells, so either a full cure time of 72 hrs was required, or cure time could be accelerated to 16 hrs when left in the high humidity of a tissue culture incubator. The choice of RTV silicone was a combination of cost considerations, ease of use, availability and reliability of bonding. We found the oxime-based curing RTV (SS-433T, Silicone Solutions, OH, USA) to be optimal because of lower viscosity which helps with evening the spread of glue on >16-well chambers. To apply the adhesive evenly and ensure a flat bonded surface, a 30mm diameter soft polymer craft roller was used to spread the glue on a 20x20cm sheet of phenolic resin or glass to generate a thin coat on the roller face, then to transfer this thin layer of glue to the chamber bonding face. Coverslips were then placed on the adhesive surface and pressed evenly with a 3D printed block (S1 Video). We prototyped using clear resin, but found light scattering too problematic for fluorescent microscopy use and switched to semi-opaque black resin. 3D printing also cannot produce optically clear plastic without polishing.

Therefore, lids for the chambers were made either by FDM printing the frames and bonding a coverslip glass using RTV silicone, by MSLA printing with a solid top for inverted fluorescence microscopy, or by MSLA printing in clear resin with a 200um thin top surface (Fig 2H). MSLA printing is preferred for the lids to allow sterilization, which is difficult in FDM layered prints. Lids were designed with a gap allowing gas exchange during tissue culture incubation.

During longer-term incubation, beyond 72 hrs, we noted a tendency of early 22x50mm chamber designs on 22x50mm coverslips to warp 0.5mm due to the absorption of humidity by the plastic. This warping was caused by a 1% expansion of the length of the plastic, but not the glass. The design solution for longer-term experiments (days to weeks) was to use a 22x22mm chamber design with a 6-chamber holder (Fig 3). This additionally allowed for greater modular flexibility to match the needs of the experimental setup in regards to the number of wells required. Another alternative was to have two linked 22x22mm chambers glued to two 22x22mm coverslips that fit within the 22x50mm holder format. For longer-term live-cell observations, we designed another holder, where single chambers fit within a Tokai HIT incubated stage (Tokai, Japan). The holder maintains the chamber at the correct height while still allowing a pressure seal to prevent $CO_2$ loss and heat escape (Fig 3). The holder takes advantage of a unique PLA formulation that is thermochromic between 32–45˚C. This allows the user a simple visual confirmation of temperature with a transition from grey to orange at 37˚C, with a second transition to yellow at 42˚C for heat shock experiments (Fig 3B–3D).

As an example of utility, a 22x22 single well chamber was used to culture live human RPE1 cells and subjected to a laser stripe DNA damage assay at 37˚C (Fig 4). After 24 hours of culture and transfection, poly(ADP-ribose) polymerase 1 (PARP1) enzyme was visualized accumulating to sites of 405nm induced DNA damage within 25 seconds by an anti-PARP1 chromobody fused to TagRFP and then visualized for an additional 2 minutes (S2 Video).

To then take advantage of this system and fast prototyping, we designed a perfusion chamber with internal microfluidic passages that could not be created using traditional subtractive

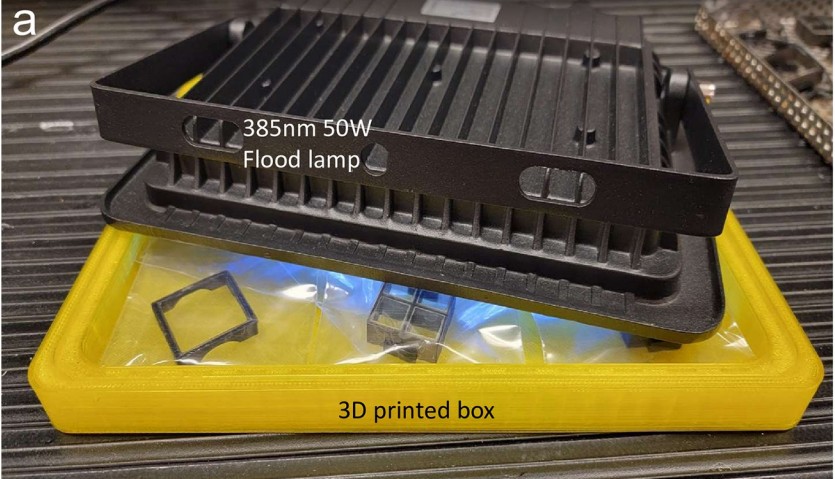
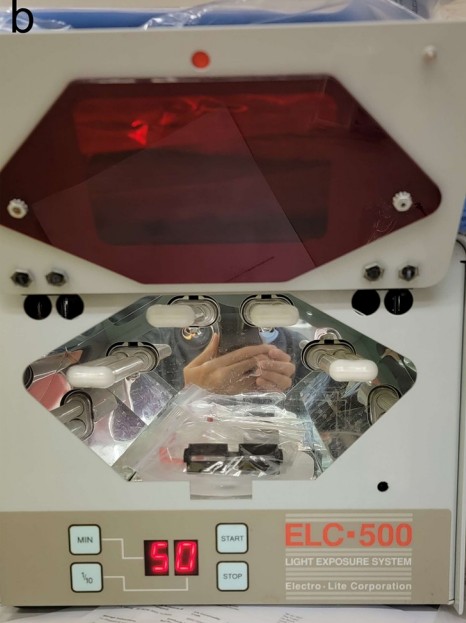

**Fig 3. Thermochromic chamber holder for live cell incubator stage.** (a) Left to right: the lid, chamber, and holder. (b) Assembly of the chamber and holder in the stage at 25˚C with grey color. (c) Placement of the chamber and holder within the incubated temperature and gas control chamber at 37˚C with color change to orange. (d) As in panel c, but temperature increased to 42˚C (yellow).

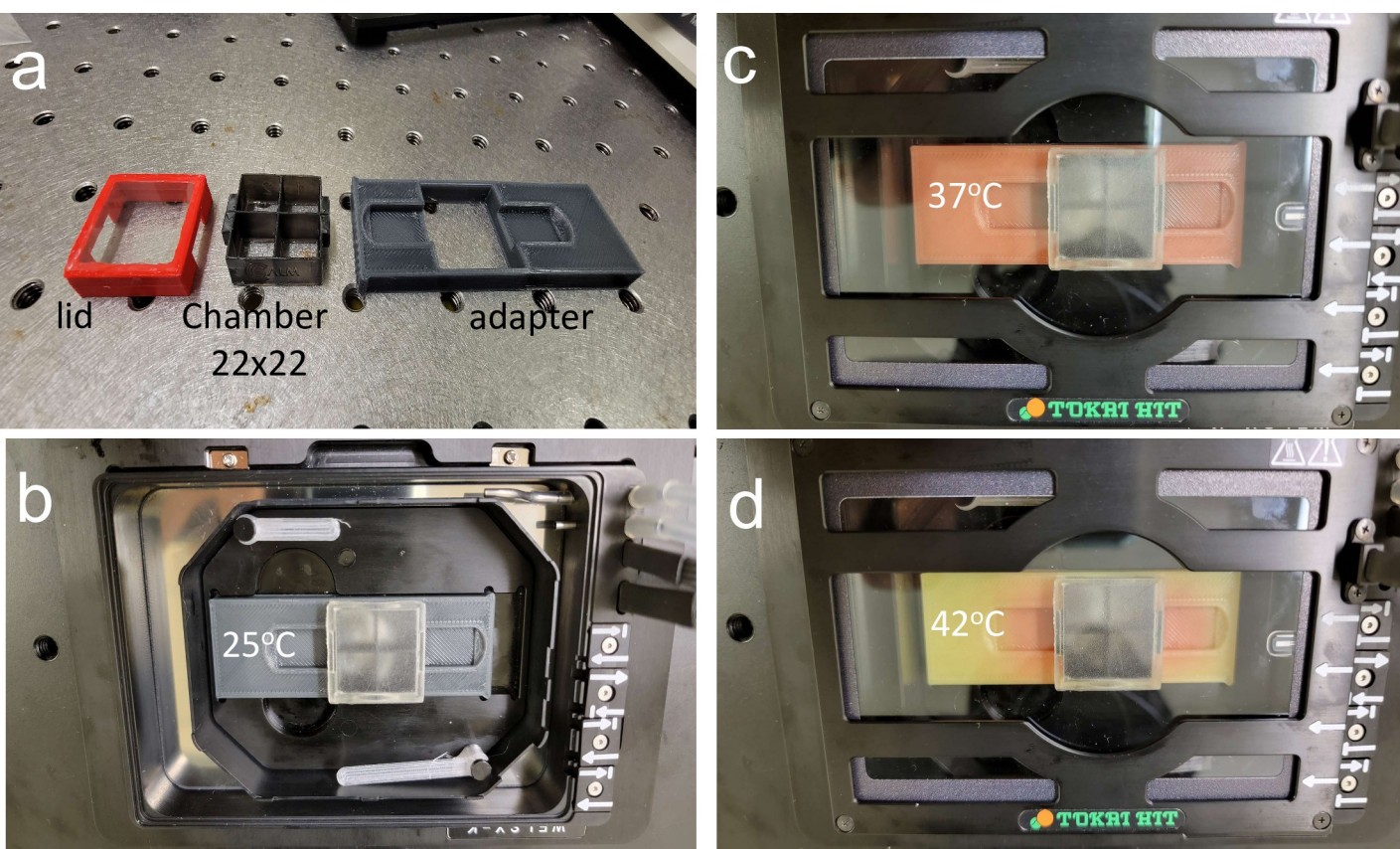

**Fig 4. Sample live cell confocal microscopy data.** RPE1 cell expressing PARP1 chromobody as red fluorescent protein fusion. (a) After 24 hrs growth, immediately after 405nm laser irradiation in the region defined by yellow box. (b) 12 seconds later. Panel c, 25 seconds later, showing PARP1 chromobody recruitment to the site of induced DNA damage. Scale bar is 10um. Timestamp is seconds and milliseconds. 60X oil objective.

manufacturing due to the encapsulated fluid channels. Since MSLA printing has excellent resolution capabilities, we were able to design a microfluidic path in a one-piece print. This model only required inserts of metal piping to connect plastic tubing to a peristaltic pump. Contrary to traditional microscopy perfusion chambers, which typically only have a single inlet and outlet, our design allowed for the even rapid distribution of fluid across the imaging area across a manifold of one into four across the surface of the 22x22mm chamber (Fig 5A). The chambers were used during tissue culture with a removable FDM printed lid bonded with silicone (Fig 5B). However, prior to completing a perfusion experiment, a 22x22mm coverslip was sealed onto the top face of the chamber using silicone vacuum grease (CS16, Silicone Solutions, OH, USA).

For sterilization, we used a combination of bathing in 70% isopropyl alcohol, bathing in sterile water, and UV irradiation in a HEPA tissue culture hood. Following the bathing steps, the chambers were allowed to dry, then sealed in clear plastic bags. The chambers were then UV sterilized in either a 365nm chamber at 30 mW/cm² (ELC-500, Electro-Lite) for 10 minutes per side, or in a bespoke 3D-printing curing/sterilization chamber. The curing/sterilization chamber was lined with aluminum foil tape and used an inexpensive (~US$40) 50W 385nm LED array flood lamp (80mW/cm²) (WOWTOU, China) (Fig 6) for at least 30 minutes per side, flipping once to avoid shadows. To ensure sterilization, previously prepared chambers from storage were irradiated with UV once more, and individual wells were washed with PBS

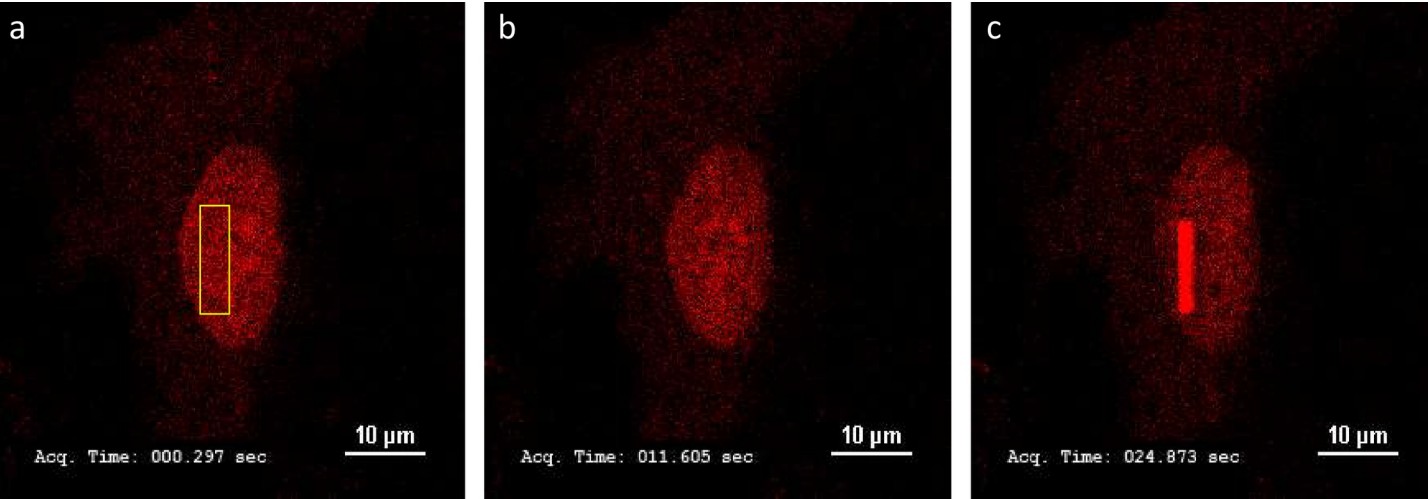

**Fig 5. Bespoke Mod3D perfusion chamber.** (a) Computer Aided Design (CAD) drawing of microfluidic paths and the 1:4 manifolds. (b) Chamber, culture lid and perfusion lid clamp. (c) A 22x22 mm coverslip is placed on top after cell culture sealed with silicone grease and clamped into place with the PETG clamp. Inlet/outlet aluminum tubing is bonded in place with RTV silicone.

and media before plating. This protocol ensured complete sterilization without any need for heat.

We did not observe any obvious growth inhibition, toxicity or morphology effects from chamber growth by TruHD cells [11] either visually or quantitatively with live/dead fluorescence assay (Fig 7A and 7B), we tested for any subtle effects on cell health by analysis of mitochondrial ellipticity, which is a very sensitive measure of cell stress [12]. Even at over 80,000 data points, no significant difference in mitochondrial ellipticity was observed versus commercial chambers (Ibidi uSlide 8 well) (Fig 7C).

The modular system resulted in significant cost savings over commercially sourced imaging chambers. 22x22mm single chambers cost 20 cents total: 2 cents in plastic and 18 cents in glass. Double chambers cost 44 cents, with the coverslips contributing to the bulk of the cost. Overall, our Mod3D solution costs were 1/25th of their commercial solution counterparts (Table 1).

## Materials and methods

The protocol described in this peer-reviewed article is published on protocols.io: dx.doi.org/10.17504/protocols.io.n92ldz587v5b/v1 and is included for printing as S1 File with this article.

### 3D printing

Models were designed by Computer Aided Design (CAD) in Fusion360 software (AutoCAD, USA) or the free web-based TinkerCAD software (https://www.tinkercad.com/). 3D models were exported as STereoLithography (.stl) files and used to generate G-code in either Cura 4.11 (Ultimaker) for FDM printing or in Chitubox freeware (https://www.chitubox.com/) for resin printing. For FDM, models were printed at 20% grid infill with a layer height of 0.16 mm on either a Creality Ender 3 or a CR10 printer (Creality, Shenzhen, China). PLA or PETG 1.75mm filament was sourced from 3D Printing Canada (https://3dprintingcanada.com/). For chamber holder tops and bottoms, printing was completed with a >6mm brim on the bottom to ensure the print remained flat. To minimize warping, the print was not removed from the heated print bed until both had cooled to room temperature. Within the print, four

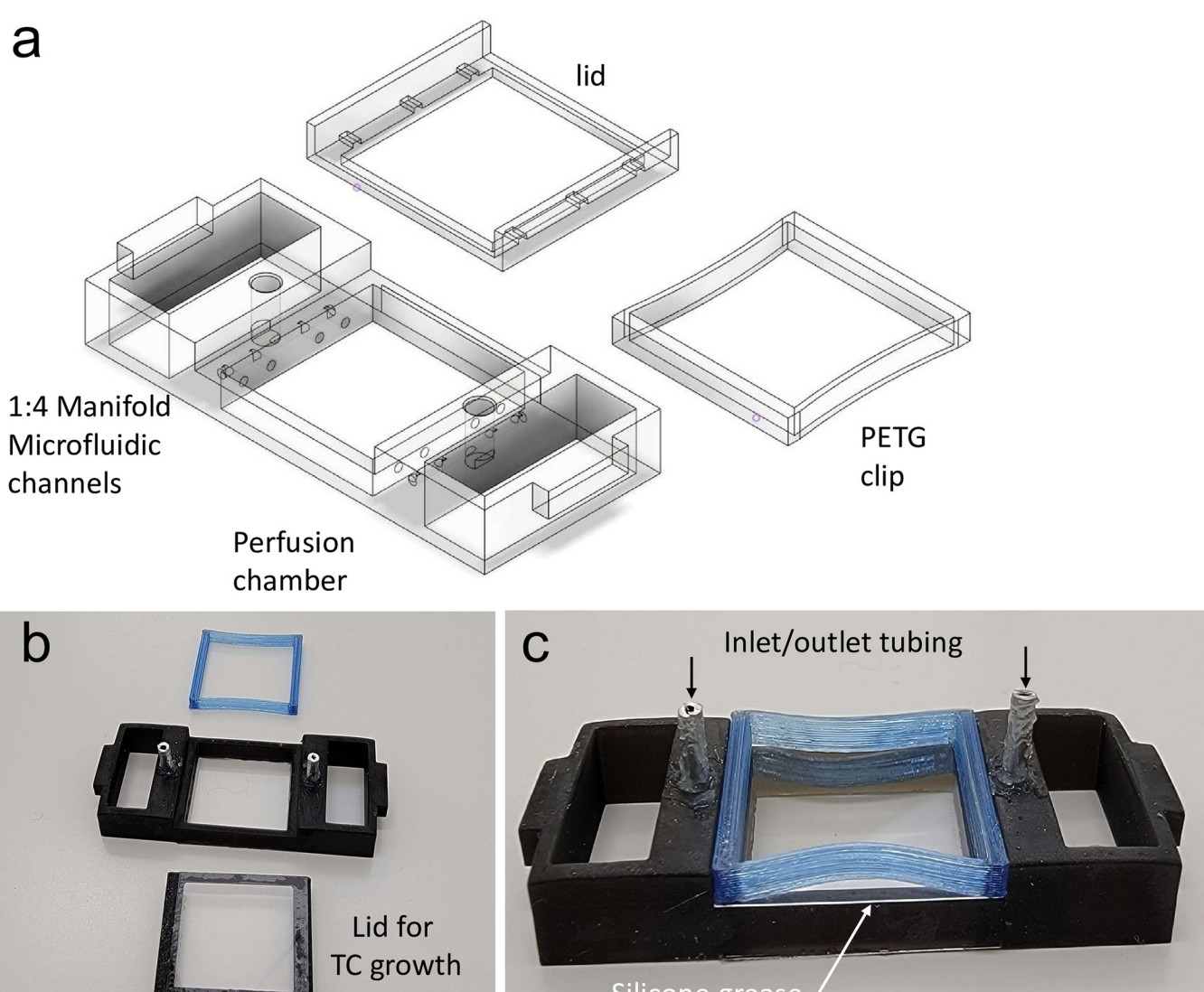

**Fig 6. Chamber UV sterilization.** (a) 3D printed curing/sterilization chamber lined with aluminum foil tape with a 30W 385 nm LED flood lamp placed on top. (b) Alternate method of chamber sterilization in a UV curing chamber with rotating platform (Electrolite ELC-500). In both methods the chambers are placed in clear polypropylene bags.

neodymium 5x1mm magnets were glued into place on each top and bottom half of the chamber holder with polyurethane or cyanoacrylate glue. Thermochromic PLA filament (TopZel Tritemp Lava) was sourced from TopZel, China.

For MSLA, an Anycubic Photon printer (Anycubic, Shenzhen, China) was used with eSun PLA BioPhotopolymer resin LC1001 (Shenzhen eSUN Industrial Co., China) using the standard settings for that printer located in the Chitubox profile, with the exception of a 160 second first later exposure and a 30 second per subsequent layer exposure. Each print used black resin and contained four 22x50mm chambers or eight 22x22mm chambers. Chambers and lids were printed directly on the printer platform without any rafts or support. Clear resin was only used for prototyping to confirm microfluidic flow.

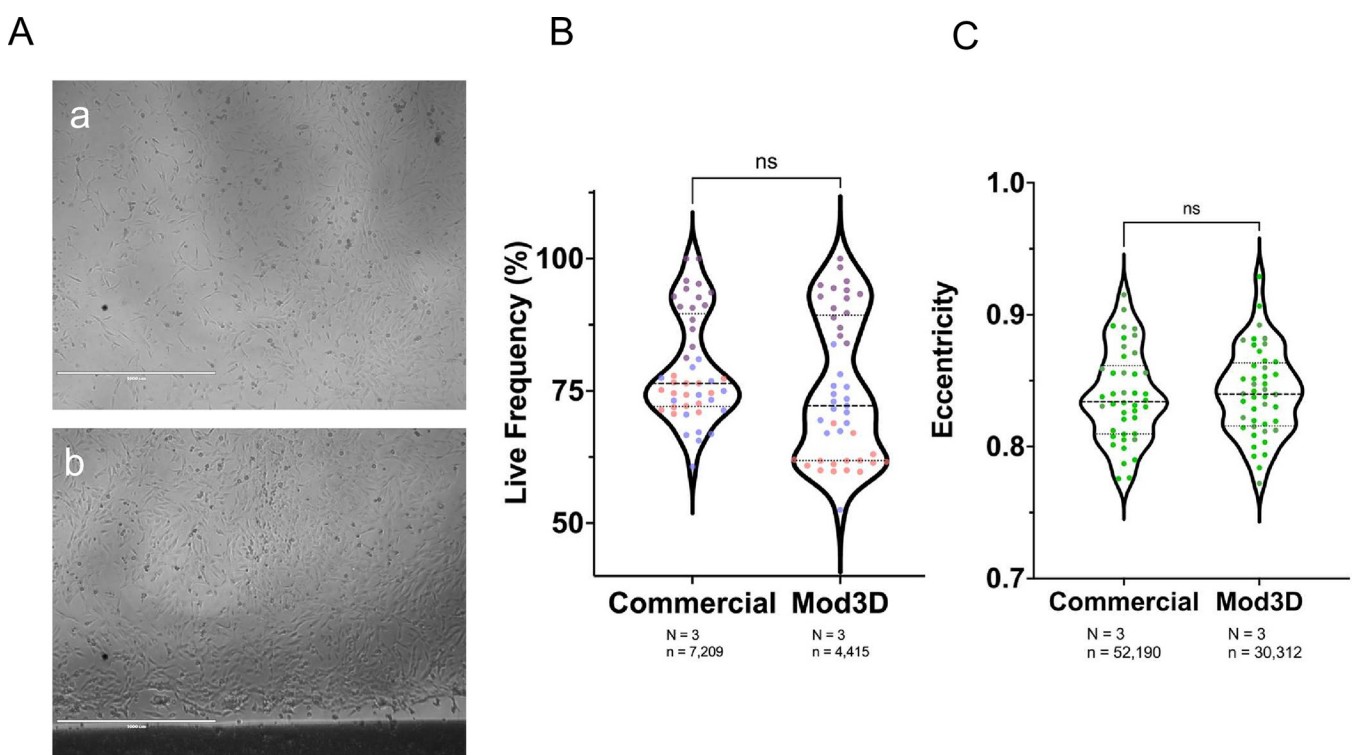

**Fig 7. Growth and no toxicity or stress in Mod3D chambers.** (A) representative bright field images of cell growth of human TruHD cells. TruHD are growth contact inhibited, non-transformed lines. Scale is 1000um. (B) Fluorescent LIVE/DEAD assay comparing commercial chambers to Mod3D. Panel C, assay of mitochondrial ellipticity using NucBlue and Mitotracker dyes comparing commercial chambers to Mod3D. No significant differences were noted, $p<0.5$ by simple random sample and a Mann-Whitney nonparametric t-Test.

Print stereolithography files (STL) are available under free use license on the NIH 3D print server: https://3dprint.nih.gov/discover/3dpx-016293.

## Post processing

After printing, 3D prints were released from the platform and washed for 15 minutes in fresh 99% isopropyl alcohol (IPA) in a Anycubic Wash & Cure Plus Machine (Anycubic, Shenzhen, China). An IPA wash in a fresh solution was important to remove any toxic monomers in the uncured resin. Chambers were then removed from the print supports prior to 405nm curing

**Table 1. Cost comparisons of commercially available imaging chambers and the 3D printed microscopy chambers.**

| Product Name | Retailer | Catalog Number | Description | Approximate Cost Per Chamber ($USD) |
|---|---|---|---|---|
| Thermo Scientific™ Nunc™ Lab-Tek™ Chamber Slide™ | ThermoFisher Scientific | 177402PK | 25x75mm 8-well polystyrene chamber on soda lime glass microscope slide | 14 |
| µ-Slide 8-Well Glass Bottom | Ibidi® | 80827 | 25.5x75.5mm 8-well plastic chamber on Schott glass slide | 12 |
| Eppendorf Cell Imaging Slide | Eppendorf | 0030742079 | 26x76mm 8-well plastic chamber on soda lime glass microscope slide | 13 |
| 8-Well Culture Slides | MatTek Life Sciences | CCS-8 | Polystyrene chamber on glass microscope slide | 9.3 |
| Mod3D live-cell Microscopy Chamber | N/A | N/A | 22x22mm/22x50mm PLA chamber on glass microscope slide | 0.20 (22x22 mm) |
| | | | | 0.55 (22x50 mm) |

in the same machine, which occurred for 30 minutes on the built-in rotating turntable. Alternatively, we used the 50W 385nm flood lamp (80mW/cm$^2$) (WOWTOU, China) placed face down on a 3D-printed box for 30 minutes, flipping once to avoid shadows.

## Chamber assembly

A 30mm diameter, 15cm wide craft roller was used to spread silicone RTV glue (SS-433T, Silicone Solutions, OH, USA) onto a sheet of phenolic resin. Once the glue was spread evenly, the roller was used to transfer it to the bottom surface of the inverted chamber. A 22x22mm #1.5 glass coverslip (VWR, USA) was then pressed onto the glue face with a 3D printed tamper block (S1 Video). The glue was allowed to fully cure for 16hrs in the humid environment of a tissue culture incubator, which accelerated the curing time for RTV silicone. Curing the chamber fully in high humidity was essential for preventing any potential toxicity from the curing agent. A similar protocol was used for the chamber lids.

## Chamber sterilization

In a HEPA tissue culture hood, both chambers and lids were bathed in 70% IPA for 10 minutes, followed by sterile water for 10 minutes, then allowed to dry under UV light. They were then placed in small clear polypropylene bags and UV irradiated prior to storage and use. UV sterilization was completed using either a 30mW/Cm$^2$ 365nm ELC-500 UV (ELC-500, Electro-Lite, USA) chamber on a rotating platform for 10 minutes per side, or the flood lamp curing chamber previously described for 30 minutes per side (Fig 6).

## DNA damage laser stripe assays

RPE1 human hTERT immortalized cells were used. Culturing conditions, 405nm laser irradiation, and microscopy equipment were as described elsewhere [13]. PARP1 enzyme was visualized with an anti-PARP1 chromobody, which comprises the anti-PARP1 VHH derived from alpaca heavy chain antibody genetically fused to TagRFP (Chromotech, Gmbh) to visualize endogenous PARP1.

## Cell toxicity and stress assays

TruHD-Q50Q40F human fibroblast cells immortalized with hTERT were cultured as described previously [11] for these experiments. Mitochondrial morphology was assessed in Mod3D and commercial plates using confocal microscopy, mitochondrial Mitotracker and nuclear NucBlue probes (Molecular Probes). Viability was assessed between Mod3D and commercial plates using microscopy of a LIVE/DEAD dye kit (Molecular Probes). Both mitochondrial and viability assays were quantified using CellProfiler (https://cellprofiler.org/), and statistically analyzed using a simple random sample and a Mann-Whitney nonparametric t-test.

# Discussion

Early experiments in prototyping these modular designs used low-cost commercial resins, which resulted in cellular toxicity. Presumably, this was due to unpolymerized monomers that did not wash out during the isopropanol washes and subsequently solubilized in media. Thus, wash steps using fresh isopropyl alcohol were essential for this protocol. Within this industry, there are over 40 different resin formulas available. However, the exact formulations are elusive and typically guarded as an industry secret. Most resins are liquid in a methyl ethyl ketone (MEK) base, contributing to a significant odor, with MEK indicated as an irritant. Therefore

all printing and handling of liquid resin were done within a chemical fume hood using gloves and goggles for handling. MEK itself is toxic and must be washed away using isopropanol. In combination with the post-printing UV curing step, isopropanol washes coincidently resulted in sterile prints.

UV resins are typically polymers of unsaturated polyester, which is an initiator and a photo-sensitizer [14]. The initiator is triggered by energy absorbed by the photosensitizer and is used to catalyze the polymerization. Typical polymers sold to hobby consumers are in the methacry-late family and are known to have toxic properties [15, 16]. However, the industry has recently evolved to produce more biocompatible polymers termed "bio inks" [17, 18]. These include polycaprolactone, PLA and polyglycolic acid (PGA) and are FDA-approved for biological use [14, 19, 20]. Of these, PLA resin (e-resin PLA, BioPhotopolymer, eSun, China) was affordable and available in our region. One unanticipated advantage to PLA plastic was the lack of surface charge seen in polystyrene products. Surface charges can lead to liquid transfer problems and have a high meniscus effect in small wells. We did not experiment above 16-wells per 22x22 chamber (or 96-wells per holder), but the limitation of the printer resolution should not be a factor for higher well densities. MSLA resin prints have a higher resistance to temperature than FDM PLA prints, and display higher flexibility with no deformation up to 100°C.

In early 22x50mm designs, we did note the expansion of chamber dimensions by up to 1%. The expansion of the chamber was due to the absorbance of moisture by the plastic within the tissue culture incubator and resulted in a 0.5mm warp on the chamber glass. This is because over three days, the plastic that was bonded to the glass expanded, while the glass coverslip did not. This warping was resolved using either a linked 22x22mm twin chamber design or by using 22x22mm chambers in a 6-chamber holder.

The modular nature of the chambers allows for efficient use of the wells based on experimental need, and therefore there is less waste from unused wells in commercially available multiwell plates. The chamber carriers, the chambers and the lid frames were made from bio-degradable PLA resin or filament [21]. The cost savings were dramatic, with a chamber costing under 30 cents, in contrast to some of the commercial examples charging over US$10. One practical aspect is the flexibility to integrate new designs. Prototypes can be generated at minimal cost in just a few hours, without the practical limitations of a local machine shop. MSLA printing time, unlike FDM, does not scale with replicates; printing one, four or eight chambers is completed at the same time. This is because the time-limiting step in MSLA is dictated by the height of the objects in the print area, not the number of objects, as all object layers are simultaneously exposed to UV mask. With FDM, print time is dictated by height, in Z, and tool movement of the extruder head, in XY. Newer MSLA printers use 4X or 8X larger screens (based on consumer computer tablet 4K or 8K LCDs), which could generate 16–32 22x50mm or 32–64 22x22mm chambers in the same 90 minute time-frame. Similarly, inexpensive consumer FDM printers on coreXY design are now available at printing speeds of up to 250mm/sec, or five times the speed of typical printers of a few years ago.

The advantages of additive local manufacturing all come into play with these designs: low cost, minimized inventory maintenance, no lag times between ordering, decreased worrying about availability and shipping costs. The assembled modular nature also means the chamber holder and lids can be reused after sterilization. The craft skills and hands-on time per step required are minimal, and a naive user can generate up to 40 chambers a day (S1 Video).

The cost of 3D printing materials and equipment is now much lower because of the expanding home consumer market. Additionally, we used a black light 50W 385nm LED flood lamp array as an inexpensive post-print processing equipment for curing and sterilization. For total equipment investment, cost recovery is realized in under 100 chambers. The 3D printing designs are freely and openly distributed on a curated website (https://3dprint.nih.gov/) (ID:

3DPX-016293) under Creative Commons license CC-BY-NC-SA. With more submissions from users for unique applications in the future, this repository should grow in designs since the cycle from concept to prototype and the ability to generate experiment-specific designs is now reduced to a negligible time and cost.

## Supporting information

**S1 Video. Gluing and assembly of Mod3D chambers to glass slides and incubation for curing.**
(MP4)

**S2 Video. Extended Live cell DNA damage laser stripe assay data from Fig 5, over 134 seconds, with timestamp.**
(MP4)

**S1 File.**
(PDF)

## Author Contributions

**Conceptualization:** Ray Truant.

**Data curation:** Siobhan Goss, Carlos Barba Bazan, Nola Begeja, Ray Truant.

**Formal analysis:** Siobhan Goss, Carlos Barba Bazan, Kaitlyn Neuman, Christina Peng, Nola Begeja, Celeste Elisabeth Suart, Ray Truant.

**Funding acquisition:** Ray Truant.

**Investigation:** Siobhan Goss, Carlos Barba Bazan, Kaitlyn Neuman, Christina Peng, Nola Begeja, Celeste Elisabeth Suart.

**Methodology:** Siobhan Goss, Carlos Barba Bazan, Kaitlyn Neuman, Christina Peng, Nola Begeja, Ray Truant.

**Project administration:** Ray Truant.

**Supervision:** Ray Truant.

**Validation:** Siobhan Goss, Carlos Barba Bazan, Christina Peng.

**Visualization:** Kaitlyn Neuman, Christina Peng.

**Writing – original draft:** Kaitlyn Neuman, Ray Truant.

**Writing – review & editing:** Siobhan Goss, Carlos Barba Bazan, Kaitlyn Neuman, Christina Peng, Nola Begeja, Celeste Elisabeth Suart.

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
