## [Decision Letter · Decision Letter 0]

23 Feb 2022

PONE-D-22-00858Mod3D: A Low-Cost, Flexible Modular System of Live-Cell Microscopy Chambers and HoldersPLOS ONE

Dear Dr. Truant,

Thank you for submitting your manuscript to PLOS ONE. After careful consideration, we feel that it has merit but does not fully meet PLOS ONE’s publication criteria as it currently stands. Therefore, we invite you to submit a revised version of the manuscript that addresses the points raised during the review process.

Please review the comments from both reviewers and make necessary edits in a revised manuscript to address their feedback.

We look forward to receiving your revised manuscript.

Kind regards,

Kristen C. Maitland, Ph.D.

Academic Editor

PLOS ONE

Journal Requirements:

2. To comply with PLOS ONE submissions requirements, please provide the Protocols.io DOI in the Methods section of the manuscript using this format: “The protocol described in this peer-reviewed article is published on protocols.io, https://dx.doi.org/10.17504/protocols.io[........] and is included for printing as supporting information file 1 with this article.” Please also provide the Protocols.io DOI in the “Protocol DOI” field of the submission form (via “Edit Submission”). For more information, please see our submission guidelines:  https://journals.plos.org/plosone/s/submission-guidelines#loc-guidelines-for-specific-study-types.

[This work was supported by NSERC discovery grant RGPIN-2020-06642 to RT.]

 [The funders had and will not have a role in study design, data collection and analysis, decision to publish, or preparation of the manuscript.]

Reviewers' comments:

Reviewer's Responses to Questions

**Comments to the Author**

1. Does the manuscript report a protocol which is of utility to the research community and adds value to the published literature?

Reviewer #1: Yes

Reviewer #2: Yes

2. Has the protocol been described in sufficient detail?

Descriptions of methods and reagents contained in the step-by-step protocol should be reported in sufficient detail for another researcher to reproduce all experiments and analyses. The protocol should describe the appropriate controls, sample sizes and replication needed to ensure that the data are robust and reproducible.

Reviewer #1: Yes

Reviewer #2: Partly

3. Does the protocol describe a validated method?

Reviewer #1: Yes

Reviewer #2: No

4. If the manuscript contains new data, have the authors made this data fully available?

Reviewer #1: Yes

Reviewer #2: Yes

**5. Is the article presented in an intelligible fashion and written in standard English?**

Reviewer #1: Yes

Reviewer #2: Yes

6. Review Comments to the Author

Reviewer #1: The paper “Mod3D: A Low-Cost, Flexible Modular System of Live-Cell Microscopy Chambers and Holders” by Gross et al provides a new method for using resin-based 3D printing to produce customized chambers for live-cell microscopy. Unlike previous papers addressing the same issue, this paper found an approach to use resin/MSLA printing to produce these chambers. This offers several advantages over filament printing used by previous studies, including higher print resolution, flatter surfaces for sealing, and faster print times. However, plastic toxicity is an issue, with a major advance of this study being identification of a specific resin and washing process which appears to eliminate these toxicity issues. In addition, a video has been provided showing the assembly process, which should ease adoption of this method by other groups as it illustrates parts of the assembly process which may not be easily communicated in text. While overall the paper is quite strong, I would recommend a few edits to strengthen the claims of non-toxicity.

Major Issues:

1. While the authors extensively describe a lack of toxicity of their chosen adhesive and resin, no direct evidence of this is provided to the reader. I would suggest including data measuring the apoptosis of cells cultured in these chambers. The classical annexin v/propidium iodide assay should be more than sufficient for this. Ideally, multiple time points should eb assessed (e.g. 24, 48 and 72 hours culture).

2. Related to issue #1, the possibility that non-lethal changes to cell behaviour may be occurring is not addressed. A measure of cell morphology, mitochondrial function, or something similar – compared to commercial chambers – would help demonstrate a lack of toxic effects on the cells.

Minor Issue:

1. While the STL files are available at the NIH exchange, no where in the body of the text is this indicated. It would be helpful if the link were provided in the methods.

Reviewer #2: This manuscript describes a protocol to print and use a microscopy chamber dedicated to "replace" disposable commercially available ones.

The chamber consists in a generic frame on which several components can be positioned, depending on the desired chamber number, shape and size.

I strongly support the idea that scientists need such openly distributed tools for making available to any lab, devices that can be expensive.

The device itself seems useful enough to deserve publication, although the manuscript needs to be amended.

I would first recommend a serious review of the existing literature about 3D-printed chambers for microscopy. There is not a single article nor a website (many STL files are freely downloadable in dedicated websites), cited in the introduction. The authors only compare their device to single-use, blister protected, sterile culture dishes dedicated to microscopy imaging.

In the introduction MSLA is mispelled, it is actually maskLESS SLA.

There is some confusion about the resin-based printers. SLA in general refers to the technique where a laser is scanned point-by-point, LCD where the light is spatially filtered with the LCD and finally DLP where the light directly comes from a projector.

In general convention as well, MaskLess SLA refers to the soft-lithography machines that are used for printing microfluidic chips, using microscopy objectives and SU8-type resins. And these machines are about several tens of kilo dollars.

The recent availability of LCD machines has made the resin-based printers affordable, but most of the SLA costs a few thousand dollars, a good DLP can reach several ten thousand dollars as well.

Could the authors screen their manuscript again and provide quantitative values (even a range) when evaluating a parameter? Such as “the resolution not up to the standards” or the “FDA superior”.

Some statements are as well blind and overestimated (unless tested): “can be reused with a sterilization cycle indefinitely”. This is most likely not possible.

And yet, sterilisation with UV in plastic bags should be carefully evaluated, since only the parts directly exposed to light could be decontaminated. Actually, the parts are very complex, and it is unlikely that the overall device could be efficiently decontaminated. Yet, biologists in general favours autoclave for decontamination of devices and parts. I am aware that the plastics and resins are not compliant with such extreme temperatures, and thus, other ways have to be found, but readers should also be warned of the risks.

I have also concerns about the “biology” section. This is indeed necessary to provide a section where the device is in use and to document the features that matter. Especially, it would be useful to provide a more detailed characterisation of the following parameters: low magnification image of a cell mat in the different type of compartments with close-ups; longer time-lapses over several hours and days to demonstrate the lack of drift (the material being softer than most of the sample holders on stages might be susceptible to bend when used in a heated environment).

Overall, I think the device is useful and could benefit to a great readership. Although since the authors would like to publish this device additionally to making it available in an open repository, the writing should be upgraded to meet the standards of research papers, including citing properly the existing literature.

7. PLOS authors have the option to publish the peer review history of their article (what does this mean?). If published, this will include your full peer review and any attached files.

Reviewer #1: No

Reviewer #2: No

---

## [Author Response · Author response to Decision Letter 0]

11 May 2022

Thank you to the reviewers for their comments and insights. Some of these comments were addressed in the detailed protocol in this submission, as this is a protocol variant of a PLOS One submission. The manuscript is meant to be read as manuscript and detailed protocol, as instructed to authors on first submission. There seems to be some concern about the validity of this protocol and what was considered before submission. For context: this was an effort over 3 years, with about 40 prototypes, 17 different adhesives and a dozen different resins. Indeed, the project was stalled for months because we could not find a non-toxic resin until PLA resins appeared in the commercial market. We also spent months on both silicone RTV and UV adhesives from multiple manufacturers, finding toxicity contrary to claims form manufacturers. We also spent months establishing a sterilization protocol that involves both chemical and UV steps, with multiple UV exposures prior to storage and before use, again, outlined in detail in the supplementary protocol. 

Reviewer #1: The paper “Mod3D: A Low-Cost, Flexible Modular System of Live-Cell Microscopy Chambers and Holders” by Gross et al provides a new method for using resin-based 3D printing to produce customized chambers for live-cell microscopy. Unlike previous papers addressing the same issue, this paper found an approach to use resin/MSLA printing to produce these chambers. This offers several advantages over filament printing used by previous studies, including higher print resolution, flatter surfaces for sealing, and faster print times. However, plastic toxicity is an issue, with a major advance of this study being identification of a specific resin and washing process which appears to eliminate these toxicity issues. In addition, a video has been provided showing the assembly process, which should ease adoption of this method by other groups as it illustrates parts of the assembly process which may not be easily communicated in text. While overall the paper is quite strong, I would recommend a few edits to strengthen the claims of non-toxicity.

Major Issues:

1. While the authors extensively describe a lack of toxicity of their chosen adhesive and resin, no direct evidence of this is provided to the reader. I would suggest including data measuring the apoptosis of cells cultured in these chambers. The classical annexin v/propidium iodide assay should be more than sufficient for this. Ideally, multiple time points should eb assessed (e.g. 24, 48 and 72 hours culture).

We did provide images of cell growth in the Supplemental Figure 1. We have now moved this into the manuscript in Figure 7A. We now include image data on cell growth, general toxicity by live/dead assay, as apoptosis assays would only assay one type of cell death (7B) and mitochondrial ellipticity with over >80,000 mitochondria assessed (7C). 

2. Related to issue #1, the possibility that non-lethal changes to cell behaviour may be occurring is not addressed. A measure of cell morphology, mitochondrial function, or something similar – compared to commercial chambers – would help demonstrate a lack of toxic effects on the cells.

This is now added as Figure 7C and is extremely sensitive given the high n values. 

Minor Issue:

1. While the STL files are available at the NIH exchange, no where in the body of the text is this indicated. It would be helpful if the link were provided in the methods.

The URL was provided at the top of the protocol (supplemental protocol) and the ID: 3DPX-016293 number was provided on line 296. We will now clarify the URL in both locations. 

Reviewer #2: This manuscript describes a protocol to print and use a microscopy chamber dedicated to "replace" disposable commercially available ones.

The chamber consists in a generic frame on which several components can be positioned, depending on the desired chamber number, shape and size.

I strongly support the idea that scientists need such openly distributed tools for making available to any lab, devices that can be expensive.

The device itself seems useful enough to deserve publication, although the manuscript needs to be amended.

I would first recommend a serious review of the existing literature about 3D-printed chambers for microscopy. There is not a single article nor a website (many STL files are freely downloadable in dedicated websites), cited in the introduction. The authors only compare their device to single-use, blister protected, sterile culture dishes dedicated to microscopy imaging.

In the introduction MSLA is mispelled, it is actually maskLESS SLA.

There is some confusion about the resin-based printers. SLA in general refers to the technique where a laser is scanned point-by-point, LCD where the light is spatially filtered with the LCD and finally DLP where the light directly comes from a projector.

In general convention as well, MaskLess SLA refers to the soft-lithography machines that are used for printing microfluidic chips, using microscopy objectives and SU8-type resins. And these machines are about several tens of kilo dollars.

The recent availability of LCD machines has made the resin-based printers affordable, but most of the SLA costs a few thousand dollars, a good DLP can reach several ten thousand dollars as well.

One point of this protocol is that consumer 3D printers have drastically reduced the costs of printing chambers like these, while depending on specialty resins and more expensive SLA printers would make the costs not worth the effort. Around 2019, low cost resin printers appeared on the market and the consumer printer industry is distinguishing them from DLP or Galvanometer-based printers by referring to them as Masked Stereo Lithography Apparatus or MSLA. This is confusing with the chip industry, which previously used the term Maskless SLA as the reviewer correctly points out. The printer we used here literally casts an LCD generated mask across a UV LED array. In the consumer 3D printing vernacular, MSLA refers to masked SLA. The following are a few typical references to this from industry:

https://www.structo3d.com/pages/mask-stereolithography-msla-technology

https://satori-tech.io/blog/msla-vs-fdm-printing-technique

https://4dfiltration.com/resources/resin-faq/lcd-masked-sla-vs-dlp-resin-printing.html

https://www.oliver3d.com/

Could the authors screen their manuscript again and provide quantitative values (even a range) when evaluating a parameter? Such as “the resolution not up to the standards” or the “FDA superior”.

Some statements are as well blind and overestimated (unless tested): “can be reused with a sterilization cycle indefinitely”. This is most likely not possible.

We had added more info on typical print resolutions for FDM versus MSLA printers, however this was in the introduction:

 “A platform controlled by a stepper motor can then provide resolutions in Z as little as 25 micrometers, with an XY resolution below 100 micrometers”

Unsure about the term “FDA superior” as it does not appear in this manuscript.

Some parts are simply holders and can be reused over and over. We have typical stage holder in use for years at this point, with a periodic 70% ethanol bath sterilization, and for stage holder, sterility is not critical. Chambers themselves are one use and disposable. 

And yet, sterilisation with UV in plastic bags should be carefully evaluated, since only the parts directly exposed to light could be decontaminated. Actually, the parts are very complex, and it is unlikely that the overall device could be efficiently decontaminated. Yet, biologists in general favours autoclave for decontamination of devices and parts. I am aware that the plastics and resins are not compliant with such extreme temperatures, and thus, other ways have to be found, but readers should also be warned of the risks.

From the detailed protocol, UV is not the sole sterilization step, nor is one step in UV alone used. There is a chemical disinfection step. By experience at this point, at >500 chambers used, we see no more contamination events than commercial gamma irradiated chambers. We will add a warning in the protocol, as the reviewer is correct that biology labs tend to heat sterilize most apparatus.

I have also concerns about the “biology” section. This is indeed necessary to provide a section where the device is in use and to document the features that matter. Especially, it would be useful to provide a more detailed characterisation of the following parameters: low magnification image of a cell mat in the different type of compartments with close-ups; longer time-lapses over several hours and days to demonstrate the lack of drift (the material being softer than most of the sample holders on stages might be susceptible to bend when used in a heated environment).

We did test the chambers for any long term warping within days of 37C incubation. This led to a change in design away from using 22x50mm slides. This was discussed in lines 266 to 271:

“In early 22x50mm designs, we did note the expansion of chamber dimensions by up to 1%. The expansion of the chamber was due to the absorbance of moisture by the plastic within the tissue culture incubator and resulted in a 0.5mm warp on the chamber glass. This is because over three days, the plastic that was bonded to the glass expanded, while the glass coverslip did not. This warping was resolved using either a linked 22x22mm twin chamber design or by using 22x22mm chambers in a 6-chamber holder.”

For stage holders, we used either PLA or PETG, which have glass transition temperatures of 50-80C, far above 37C incubation conditions. With new references added, we should not need to re-establish the stability of these plastics for microscopy.

We have moved cell imaging data from supplemental Figure 1, in addition to the laser stripe experiment in the original version, plus a high data number measure of any mitochondrial stress, as suggested by both reviewers, as well as a general live/dead assay comparing Mod3D to commercial chambers. This is now in Figure 7. Both reviewers raised this point and we agree this is important to validate the utility of the chambers and this justifies addition to the manuscript as Figure 7. Note at even very high data point counts >80,000, we could not see differences in mitochondrial ellipticity, even in these non-transformed cell lines that are very sensitive to stress being derived from Huntington’s disease patients. 

Overall, I think the device is useful and could benefit to a great readership. Although since the authors would like to publish this device additionally to making it available in an open repository, the writing should be upgraded to meet the standards of research papers, including citing properly the existing literature.

We had a very difficult time searching the literature for consumable microscopy chambers. We found a chamber system for zebrafish models, and molds for silicone chambers, these references are now added as well as some broader references on the use of 3D printing on open sourced projects. We added 9 additional references to the revised manuscript.

---

## [Editor Report · Decision Letter 1]

19 May 2022

Mod3D: A Low-Cost, Flexible Modular System of Live-Cell Microscopy Chambers and Holders

PONE-D-22-00858R1

Dear Dr. Truant,

We’re pleased to inform you that your manuscript has been judged scientifically suitable for publication and will be formally accepted for publication once it meets all outstanding technical requirements.

Kind regards,

Kristen C. Maitland, Ph.D.

Academic Editor

PLOS ONE
---

## [Editor Report · Acceptance letter]

25 May 2022

PONE-D-22-00858R1 

Mod3D: A Low-Cost, Flexible Modular System of Live-Cell Microscopy Chambers and Holders 

Dear Dr. Truant:

I'm pleased to inform you that your manuscript has been deemed suitable for publication in PLOS ONE. Congratulations! Your manuscript is now with our production department. 

Kind regards, 

on behalf of

Dr. Kristen C. Maitland 

Academic Editor

PLOS ONE